# Antibody Identification for Antigen Detection in Formalin-Fixed Paraffin-Embedded Tissue Using Phage Display and Naïve Libraries

**DOI:** 10.3390/antib10010004

**Published:** 2021-01-14

**Authors:** Célestine Mairaville, Pierre Martineau

**Affiliations:** IRCM, Institut de Recherche en Cancérologie de Montpellier, INSERM U1194, Université de Montpellier, F-34298 Montpellier, France; celestine.mairaville@inserm.fr

**Keywords:** immunohistochemistry, phage display, monoclonal antibody

## Abstract

Immunohistochemistry is a widely used technique for research and diagnostic purposes that relies on the recognition by antibodies of antigens expressed in tissues. However, tissue processing and particularly formalin fixation affect the conformation of these antigens through the formation of methylene bridges. Although antigen retrieval techniques can partially restore antigen immunoreactivity, it is difficult to identify antibodies that can recognize their target especially in formalin-fixed paraffin-embedded tissues. Most of the antibodies currently used in immunohistochemistry have been obtained by animal immunization; however, in vitro display techniques represent alternative strategies that have not been fully explored yet. This review provides an overview of phage display-based antibody selections using naïve antibody libraries on various supports (fixed cells, dissociated tissues, tissue fragments, and tissue sections) that have led to the identification of antibodies suitable for immunohistochemistry.

## 1. Introduction

Immunohistochemistry (IHC) is a procedure routinely used for research and diagnostic purposes [1] that relies on the detection by antibodies of antigens in tissue sections. Depending on the revelation system, tissue staining is monitored by bright-field or fluorescence microscopy [2], the latter being referred to as immunofluorescence (IF). Staining can be performed on frozen sections (IHC-f/IF-f) and on formalin-fixed paraffin-embedded (FFPE) sections (IHC-p/IF-p). Tissue processing strongly affects the reactivity of antibodies, and the same antibody is rarely efficient on both types of tissue sections [3,4]. This difference is mainly due to formalin fixation that affects antigenicity of the targeted antigen. The aim of the fixation step is to prevent tissue autolysis, putrefaction, and degradation, and to preserve the histomorphology and the subcellular localization of cellular components [5,6], but no fixative fully fulfills these different aims [7].

Freezing is widely used in immunohistochemistry because it is associated with good antigenicity, superior molecular preservation, and shorter processing times than FFPE samples [4]. Yet, formalin fixation followed by paraffin embedding remains the standard procedure for tissue preservation [8,9]: it guarantees the good conservation of the histological morphology, pathologists have become accustomed to the artifacts caused by this type of fixation and storing the blocks obtained is easier and cheaper than for frozen tissues [8,9,10]. The lower antigenicity of FFPE samples is mainly explained by the reactivity of formaldehyde, present in formalin, with several chemical groups found in amino acids, particularly through the Mannich reaction, the formation of methylol adducts or Schiff bases [11]. Ultimately, these reactions can lead to the formation of methylene bridges that modify protein conformation and epitopes, resulting in poor antibody reactivity. This phenomenon is known as antigen masking. To restore the immunoreactivity of the fixed antigens, antigen retrieval techniques that involve the use of proteases (trypsin, proteinase K) or heat have been developed [12]. The contribution of heat-induced epitope retrieval methods to immunohistochemistry is so important that experts distinguish between the time before and after their introduction [9,13,14]. Nevertheless, these techniques remain empirical and the underlying mechanisms are not fully understood [15]. Moreover, they must be optimized for each antibody-antigen couple [2], and some antigens do not fully recover their native reactivity even after epitope retrieval. Therefore, it may be difficult to identify antibodies that can bind to such antigens. This means that a large collection of antibodies must be screened, and success is not always guaranteed. For instance, Morimoto’s group tested 13 home-made antibodies and 26 commercial monoclonal antibodies against CD26 without identifying a reliable monoclonal antibody for IHC [16,17].

Traditionally, the monoclonal and polyclonal antibodies used for IHC/IF are produced by immunizing animals with the targeted protein or peptides [7]. The advantages and drawbacks of these antibodies have been summarized by Ramos-Vara and Miller [7] and Uhlén and Pontén [18]. However, to generate antibodies that are difficult or impossible to obtain with such techniques and to improve animal welfare, alternative methods are needed and may even become mandatory in the future [19,20].

Due to its flexibility and robustness, phage display is the most widely used technique for in vitro selection of antibodies [21,22]. Phage display-based antibody selection is generally performed on purified antigens or on cells [23,24]. However, in culture, cell lines undergo genetic drift, resulting in phenotypic changes, such as variations in the expression of membrane proteins [25,26]. Differences with cells in vivo are also favored by the absence in cultured cells of microenvironmental stimuli through cell–cell or cell–matrix interactions and soluble factors [27,28]. Moreover, IHC can be used also to investigate the complex mixture of cell types and molecules of the tissue microenvironment. For instance, in cancer, cancer-associated fibroblasts, immune cells, blood and lymphatic vessels, as well as matrix and extracellular elements represent a collection of potential antibody targets that are missed when using cultured cell lines for phage display-based antibody selection [29]. Tissue sections and tissue fragments allow targeting both cellular and histological abnormalities, and thus constitute a complex but clinically relevant support for antibody selection by phage display. Obtaining fresh tissues can be difficult, but frozen and FFPE tissue sections can also serve as material for antibody selection, as discussed in this review. Nevertheless, as reminded by Pimenidou et al., “in most cases, the isolated phage antibodies will only bind to tissue presented in a similar form, i.e., antibodies selected on frozen tissue recognize fresh or frozen tissue, but not formalin-fixed paraffin-embedded tissue, and vice versa” [30].

Several recent reviews have focused on the different types of antibody libraries, particularly the recently described naïve libraries based on human or synthetic repertoires [31]. Such libraries are at the basis of the experiments described in this review, but any of them could be used and, therefore, they will not be discussed here. Although we will focus here on antibodies, alternative probes can be developed using the same strategies. In particular, several publications have described the use of phage-displayed peptide libraries, panned on fresh tissues, dissociated or not [32,33,34,35,36,37], and on tissue sections [38,39,40], eventually associated with laser capture microdissection. In these publications, the peptides were however rarely validated by IHC or IF.

In a recent review, Sánchez-Martín et al. described phage display-based strategies on tissue and live animals to identify new targets for cancer diagnosis and treatment [41]. Although there is some overlap with their review, the present article will focus on the antibodies and not the targets, and specifically on strategies based on naïve antibody libraries and phage display to select antibodies that can recognize the target in tissues with an emphasis on FFPE-compatible antibodies. In the field of protein evolution, including phage display approaches, it is noteworthy that “you get what you select (screen) for” [42]. As explained above, tissue fixation introduces a strong constraint on protein antigenicity and therefore on antibody selection, and it would thus be wise to use FFPE tissue for the whole phage display process. This is however not very practical and substitutes are frequently used, in particular, during the initial screen that evaluates a large number of binders. This review will focus on selection strategies that use FFPE tissues for panning or sensible substitutes, like fresh or fixed dissociated tissues, tissue fragments or fixed cells. This is summarized in Figure 1 and Table 1. On the contrary, the use of FFPE sections or related antigens during the panning step is not needed if only frozen sections have to be analyzed, as demonstrated by the high-throughput isolation of IHC-f compatible antibodies using classical phage display on purified proteins [43].

## 2. Antibody Selection on Processed Cells and Tissues

### 2.1. Antibody Selection on Fixed Cells

Phage display-based antibody selection is frequently performed on cell lines because this support allows the correct folding of the target protein and the identification of antibodies against difficult-to-purify multiple transmembrane proteins. Gur et al. described a strategy based on the SUM159 cell line, a model of breast cancer stem cells [44] (Figure 1). In culture, this cell line contains a small fraction of cancer stem cells (5.6%) that can be identified by their higher aldehyde dehydrogenase 1 (ALDH1) activity. Gur et al. selected phages specific for the ALDH1^+^ population by five rounds of negative and positive panning using the sorted ALDH1^−^ and ALDH1^+^ cells as bait, respectively. Unlike classical on-cell selections, they performed panning on cells fixed with 4% paraformaldehyde (PFA). After the fifth panning round, they tested 171 individual phages by immunocytochemistry (ICC) on fixed cells and identified two clones specific of ALDH1^+^ cells. These clones recognized their target in FFPE sections of SUM159 cell pellets and in frozen SUM159 cell xenograft tissue sections, with a stronger binding to ALDH1^+^ than ALDH1^−^ cells. They performed all experiments with phage particles that display single chain fragment variable (scFv) fragments, and never confirmed their results with soluble purified scFv or IgG molecules. Finally, they did not demonstrate the specificity of their clones for ALDH1^+^ cells in mouse xenograft tissue sections, and did not study the binding in non-SUM159-derived tissues and in other cell types.

### 2.2. Antibody Selection on Dissociated Tissues

The main limitation of using cell lines is that they are a poor surrogate of normal and pathologic tissues. Particularly, cell–cell contacts and tissue heterogeneity are lost, resulting in the modification of the accessibility and expression level of many proteins [25,26,27,28]. This prompted many groups to perform phage display-based antibody selections, ex vivo, on tissues freshly removed from patients by surgical resection or more rarely from a (tumor-bearing) mouse [41,45]. Tissue samples are generally dissected, rinsed, and minced into small pieces with a razor. Tissue fragments or dissociated cells are then used for antibody selection. Tissue dissociation is frequently achieved by enzymatic digestion, most often with collagenase [45,46,47] or hyaluronidase [46] (Table 1, column “Support” for Selection). Dissociated cells are then directly used for panning or after fixation with PFA [48,49] using naïve or immunized libraries. Finally, antibody selection is done against the cell pool or a specific cell type [45,49] captured with an antibody against a specific marker (Figure 1).

#### 2.2.1. Antibody Selection on Freshly Dissociated Tissues

Jakobsen et al. performed, without depletion, two rounds of panning on freshly isolated cells obtained by dissociation of a breast cancer surgical sample [46]. Screening of the selected phages by ELISA on a collection of fixed cancer cell types, followed by IHC-p, led to the identification of two tumor-binding clones (Ab39 and Ab83 during the first and second round, respectively). Ab39 seems more specific for cancer tissues than Ab83, and targets glucose related protein 78 (GRP78; Table 2). GRP78 is an intracellular protein in normal cells, but in many tumors, it is overexpressed and partially relocates at the cell surface. Edwards et al. performed seven selections in parallel, among which three on dissociated and two on sliced adipose tissue samples [47]. The massive screening of more than 3000 clones by ELISA on immobilized plasma membranes indicated that 961 phages (32%), which represented 200 different scFv sequences, displayed some degree of specificity (3 times above the background). A wider screen on adipose tissue samples and five other cell types showed that 109 phages were specific for the adipocyte plasma membrane. Among these 109 unique specific clones, 80% were selected using the approach on dissociated tissue samples. The 109 clones were evaluated by phage-IHC on frozen sections: 82 clones generated 50 different staining patterns, and all recognized adipocytes in up to 37 tissues, but also stained at least another cell type. However, in these two studies [46,47], the authors never evaluated the binding of soluble scFv or IgG, but only of phage-scFv, a format that cannot be routinely used by histology laboratories.

#### 2.2.2. Antibody Selection on Fixed Dissociated Tissues

To identify tumor-associated antigens for colorectal cancer immunotargeting, Roovers et al. performed antibody selection on cells originating from tumor tissue specimens dissociated with EDTA, EGTA, and dithiothreitol (DTT) [48]. Cells were fixed with 0.25% PFA at 4 °C for 20 min before selection. After five rounds of selection against tumor cells obtained from a different patient for each round, soluble Fab fragments were tested by IHC-f on fixed cryosections. Only three Fab fragments from the fifth selection round (and none from the fourth) gave a positive staining. Two clones (B5 and C8) stained exclusively the malignant tissue, and the third Fab (B8) recognized stromal cells. Clones B5 and C8 were tested by flow cytometry using fixed and unfixed colorectal cancer cell lines. The two Fab fragments recognized tumor cells only after fixation, and epitope binding was abolished by DTT treatment. This suggests that these antibodies recognize an epitope modified by fixation with PFA of the cells used for panning. In addition, although the data were not shown, the authors mentioned that clone C8, and not the other two, could be used with FFPE sections.

#### 2.2.3. Antibody Selection against Specific Cell Subtypes

When tissue samples are chosen for panning, several cell types are present and it is difficult to specifically target the desired cell type. To overcome this problem without additional cell purification steps, a cell marker may be used to capture the targeted cell type and select specific phages in a single step.

To identify in the thymus microenvironment surface epithelial markers that could be implicated in T-cell maturation, Palmer et al. incubated dissociated mouse thymus cells with a phage display antibody library [45]. The targeted epithelial cells, which express MHC class II proteins, were captured thanks to an antibody against this complex coupled with magnetic beads. As the MHC class II+ cells represented a small proportion (2%) of the total cell population, they hypothesized that only specific phages would be retained by the positive cells because the non-specific phages would be mainly captured by the negative cells. The selection led to the identification of several scFv fragments that recognized in situ the cortical epithelium or/and a subset of medullary cells by IHC on thymus cryosections. However, the authors did not demonstrate their specificity on other tissues or using soluble scFv or IgG. To target the tumor vasculature, Mutuberria et al. chose a similar magnetic sorting approach using MACS columns and an anti-CD31 antibody to capture human umbilical vein endothelial cells (HUVECs) freshly harvested from normal umbilical cords and previously incubated with a naïve human Fab-fragment phage display library [49]. After 3–4 rounds of selection, phages were first screened by sequencing and then by flow cytometry on HUVEC. The 15 clones that were positive by fluorescence-activated cell sorting (FACS) were then screened by IHC-f, and 11 could stain tumor sections. Four clones that gave a particularly strong signal were further characterized and showed high specificity toward different cancer tissues versus normal tissues. However, their specificity was not confirmed using purified Fab or IgG.

### 2.3. Antibody Selection on Tissue Fragments

Tissue dissociation may affect protein accessibility or the structures present only in tissues. Therefore, it would be wiser to directly select antibodies on tissue fragments without any enzymatic or mechanical disruption. Several studies showed that this approach is feasible using intact tissue pieces [50], after mincing with a razor blade or scissors [47,51,52], or after microtome sectioning [53]. Initially or after processing, tissues can be fixed with PFA [51,52,53] to better mimic the IHC-p conditions (Figure 1).

#### 2.3.1. Antibody Selection on Fresh Tissues

In addition to dissociated adipose tissues, Edwards et al. [47] tested phage selection also on tissue fragments. When selection was performed at 4 °C on tissue fragments originating from the same location (abdominal, subcutaneous) as the dissociated samples, and the same selection rounds were considered (2 and 3), the results of phage-ELISA on immobilized plasma membranes were similar: 454/1208 (37.6%) and 64/190 (33.7%) positive clones on dissociated tissues and on tissue fragments, respectively. Nevertheless, when the 200 unique positive clones from the seven selection rounds were screened by phage-ELISA on membranes of additional and different cell types, three times more positive and specific phages were obtained from the selections on dissociated tissues (87/2242; 3.9%) than on tissue fragments (5/380; 1.3%). Unfortunately, the authors did not specify from which selection (sliced or dissociated tissue) originated the 82 phage clones tested by IHC-f. Dorfmueller et al. performed selections on corneal tissue and on monolayers of human corneal endothelial cells (hCEC) in microfluidic chambers to identify antibodies against these cells [50]. The first round of selection on tissue (C1) was followed by two additional rounds on tissue (C2 and C3) or by two-three rounds on cells (C1M2 or C1M3). A negative selection step on primary fibroblast cultures was performed before all selection rounds and also after selection on tissue. Finally, 1248 soluble scFv from the C3, C1M2, and C1M3 rounds were assessed by ELISA, yielding an average positive rate of 5.8% on fixed hCEC. Three sequences were found multiple times (7 times for two sequences and 48 times for one sequence) among the 79 positive clones. These three scFv sequences were cloned and expressed as alkaline-phosphatase-scFv fusion proteins, and then tested by ICC and IHC-f. The two clones obtained during the C3 round displayed a non-specific staining of fibroblasts and were discarded. The third scFv fragment, from the C1M2 round, was specific for hCEC. When expressed in the IgG format, this antibody was still specific for corneal endothelium although fibroblast staining was increased, because the target, identified as ALCAM (CD166) (Table 2), is also expressed by these cells at lower level.

#### 2.3.2. Antibody Selection on Fixed Tissues

Van Ewijk et al. performed phage display library panning on thymus fragments [51,52]. Tissues were fixed by glutaraldehyde perfusion before removal to prevent phage internalization during overnight panning. The four rounds of panning were performed on thymus fragments from mouse strains with different MHC haplotypes. Individual phage clones were then directly tested by IHC-f and gave various staining patterns of epithelial cells in cortex and medulla. For example, clone TB4-20 stained all cortical and medullary epithelial reticular cells. This clone, and to a lesser extent clone TB4-4, showed cross-reactivity toward human thymus. Although the depletion step on thymocytes and splenocytes effectively limited the selection of antibodies against lymphoid cells, the selected clones were not thymus-specific and stained several other epithelial tissues. Jarutat et al. used an original approach in which selection was performed on “free-floating” FFPE sections [53]. Free-floating sections in a tube were deparaffinized and underwent epitope retrieval, like for the classical IHC techniques, before panning. Antibody selection was performed on FFPE mantle cell lymphoma tissue sections after library packaging with Hyperphage to increase Fab display and multivalency. Except for the first round, the phage stocks were depleted on normal tonsil tissue prior to selection. After six selection rounds, 240 clones randomly picked from rounds 4 to 6 were directly tested by IHC-p, with hit rates of 3%, 25%, and 98%, respectively. This shows a strong enrichment of phages that can recognize their antigen in situ. Moreover, most of these clones gave a strong signal. Overall, five different staining patterns were observed, although most of them were not specific for the pathological tissue. The most enriched clone from round 6 (AbyD02701) generated a staining pattern that concerned malignant tissue and also normal mantle and interfollicular cells. As mantle cell lymphomas derive from these cells, this clone was further characterized and its target was found to be vimentin (Table 2). The antigens recognized by the other clones were not investigated.

## 3. On-Slide Antibody Selection

Antibody selection carried out on tissue sections mounted on glass slides is particularly interesting for the identification of IHC-specific antibodies because such tissue samples are prepared as for IHC staining and therefore, antigen accessibility, conformation, and chemical modifications are comparable. In practice, this antibody selection strategy presents strong similarities with the usual IHC staining procedures. Accordingly, many parameters may differ during antibody selection, particularly the type of tissue sections (frozen or FFPE), their thickness (2–8 µm), potential fixatives (acetone, PFA +/‒ methanol), experimental conditions (concentration, time, temperature), and epitope retrieval conditions (none, heat-based at pH 6 or 9, or protease-based). Before fixation, it is recommended to inspect the tissue samples and remove irrelevant components, such as fat or necrotic tissues, because they may result in freezing heterogeneity [54]. Most tissues are heterogeneous, and sometimes it is desirable to target only a particular cell type or structure. Approaches using capture antibodies and dissociated cells have been described in Section 2.2.3. This section will present selection strategies performed on tissue sections using a first marker or histologic criteria to specifically target one or several regions of interest (ROI).

Antibody selection performed on whole frozen sections with immune libraries [54,55,56] will not be discussed here because this review is focused on naïve libraries. The present section describes antibody selection on whole FFPE sections [57,58], and also three on-slide micro-selection strategies to target specific tissue structures or cell clusters (Figure 1). Two of them rely on the excision of the positive tissue by laser-assisted microdissection [59,60,61,62,63] or with a micropipette [64]; the third one blocks bacterial replication of irrelevant phages through UV-induced DNA damage [65,66,67,68,69,70,71]. All these strategies can be used on FFPE and frozen tissue sections. With the exception of the studies by ten Haaf et al. [57,58] and by Sun et al. [62], the antibody selection strategies described here include a phage display library panning step directly on slides where the rare cells of interest are surrounded by a large excess of other cells (Table 1, “Depletion”). The presence of these different cell types on the slides allows the simultaneous selection and depletion by competition. Indeed, phages that recognize antigens present on the cells of interest and on other cells will preferentially bind to the latter because of their abundance.

### 3.1. On Whole FFPE Sections

To target cancer cells in their microenvironment, ten Haaf et al. selected them directly on slides using lung cancer FFPE tissue sections from different patients and healthy lung tissue samples for the depletion step [57,58]. Each of the three rounds of panning was carried out on four slides, to test different epitope retrieval conditions. Then, 440 phage clones were screened by ELISA on immobilized membrane fragments, leading to the identification of 207 positive clones. Again, several epitope retrieval conditions were compared before evaluating the clones by IHC-p. Among the nine clones further tested as soluble Fab fragments, three were specific for lung cancer tissue, without staining on normal lung tissue and no or minimal cross-reactivity with other healthy tissues. The in vitro assays and the first screening steps showed that these three clones also recognized the native form of their unknown targets. When coupled to a toxin, two of these clones induced cytotoxicity in a dose-dependent manner (half maximal inhibitory concentration of 21–23 nM) and exclusively in lung cancer cell lines.

### 3.2. Laser-Assisted Microdissection Strategies

Laser capture microdissection (LCM), developed by Emmert-Buck et al., was first performed by disposing a thermoplastic film above tissue sections (frozen or FFPE) or cytological preparations on glass slides [72]. The film is then melted by an infrared laser source, thereby embedding in the film the underlying tissue/cells of interest and allowing their capture. To simplify the transfer, the film is directly bound to a vial cap [73]. Concomitantly, Schütze and Lahr developed a technique based on ultraviolet rays. A focused laser beam excises the ROI (i.e., laser microbeam microdissection) that is then catapulted into a collecting tube thanks to a defocused laser beam (i.e., laser pressure capture) [74,75]. These laser-assisted microdissection (LMD) techniques and their modifications make tissue excision faster and less manipulator-dependent than manual microdissection. LMD has been exploited for selection by phage display.

**Table 1 antibodies-10-00004-t001:** Phage display-based antibody selection for tissue characterization.

Authors and References	Antibody Selection					Screening					
	Selection			Depletion		First Screening		Immunohistochemical Staining			
	Nb of Rounds	Support	Fixation	Before, during or after Selection	Support	Nb of Clones	Technique	Type	Antibody Format	Positive Clones	Note
**Selections on processed cells and tissues**											
Gur et al. [44]	5	Cell lines	4% PFA	Before	Negatively-sorted cells	171	Phage-ICC on fixed cells	IHC-p/IHC-f	Phage-scFv	2/2	Both clones stain more intensely ALDH1+ than ALDH1- cells
Edwards et al. [47]	3	Tissue samples dissociated with collagenase	/	/	/	2242	Phage-ELISA on cell membranes	IHC-f	Phage-scFv	82/109	All cross-reacted with at least another cell type or structure
Jakobsen et al. [46]	1–2	Tissue samples dissociated with collagenase + hyaluronidase	/	/	/	Probably ^b^ between 83 and 98	Phage-ELISA on fixed cells	IHC-p	Phage-scFv	2/2	Reactive with tumors of different histologic origins; no or weak binding to normal tissues
Roovers et al. [48]	5	Tissue samples dissociated with EDTA, EGTA, DTT	0.25% PFA, 4 °C, 20 min	/	/	42 clones with distinct fingerprint pattern tested	IHC	IHC-f	Fab	3/40	Only one clone can stain FFPE sections
Mutuberria et al. [49]	3–4	Tissue samples dissociated with trypsin, EDTA and cultured before selection	1% PFA, RT, 30 min	During	Cells, magnetic sorting	132 clones fingerprinted, 17 unique clones tested	Flow cytometry	IHC-f	Phage-scFv	11/17	/
Palmer et al. [45]	6	Tissue samples dissociated with collagenase	/	During	Cells, magnetic sorting	At least ^b^ 85	Phage-IHC	IHC-f	Phage-scFv	7 ^c^	None stained exclusively all medullary epithelium
Edwards et al. [47]	3	Non-dissociated tissue fragments	/	/	/	380	Phage-ELISA on cell membranes	IHC-f	Phage-scFv	82/109	All cross-reacted with at least another cell type or structure
Dorfmueller et al. [50]	3–4	Non-dissociated tissue samples	/	Before and after	Cells (primary culture)	1248	On-cell ELISA	IF-f	scFv-alcaline phosphatase fusion proteins, then IgG	at least 4/20	Number of non-specific clones not mentioned.
Jarutat et al. [53]	6	Free-floating FFPE sections	FFPE	Before (only from the 2nd round)	Healthy tissue sections	240	IHC	IHC-p	bacterial lysates containing Fab or mini-antibodies	74/240	Up to 6 clones tested per slide.
Van Ewijk et al. & Radošević et al. [51,52]	3–4	Non-dissociated tissue fragments	Glutaraldehyde ^a^	Before and simultaneously	Cells (thymocytes and fixed spleen cells)	Probably ^b^ at least 28	Phage-IHC	IHC-f	Phage-scFv then scFv	3 ^c^	/
**On-slide selections**											
ten Haaf et al. [57,58]	3	FFPE sections on slides	FFPE	Before	Healthy tissue sections	440	Phage-ELISA on cell membranes	IHC-p	Fab	3/3	No or minimal cross-reactivity toward healthy tissues
Ruan et al. & Su et al. [60,61]	2	Cryosections on slides, with LMD	/	During	Rest of the slide	192	Flow cytometry	IHC-p/IHC-f	biotinylated-scFv	1/1	Clone can stain only cryosections; cross-reactive with some healthy tissues.
	2	FFPE sections on slides, with LMD	FFPE	During	Rest of the slide	760	Flow cytometry	IHC-p/IHC-f	biotinylated-scFv	1/1	Clone can stain FFPE and cryosections; low cross-reactivity with healthy tissues.
Tanaka et al. [59]	1–2	Cryosections on slides, with LMD	Acetone, 5 min	/	/	409 PCR-controlled clones; 157 unique clones tested	Phage-IHC	IHC-f	Phage-scFv	5/9	/
Sun et al. [62]	1	Catapulted cryosections, with LMD	2% PFA, RT, 15 min, or FFPE	/	/	79, all unique	IF-f	IF-f	Phage-scFv	>14/79	14/79 bound to cancer cells more intensely than to tumor stroma
Sun et al. [63]	1–3	Cryosections on slides, with LMD	2% PFA, RT, 15 min	/	/	150	IF-f	IF-f/ IHC-p	Phage-scFv	31/150 and 6/150	Selection of a patient-specific clone
Sørensen et al. [66,67]	1	Cytological preparations, with shadow stick	Methanol + PFA	During	Rest of the slide (male cells)	1536	On-cell phage-ELISA	/	/	/	/
Sørensen et al. [68]	1	Cytological preparations, with shadow stick	/	During	Rest of the slide (female cells)	12 clones; 10 tested	ICC	IF-p	scFv	5/10	/
Larsen et al. [65]	1	FFPE sections, with shadow stick	FFPE	During	Rest of the slide	40	On-cell phage-ELISA	IHC-p	scFv	2/3	Clone 2E confirms the feasibility of shadow stick selections on tissue
Larsen et al. [69]	1	Cryosections, with shadow stick	PFA, 10 min	During	Rest of the slide	315	On-cell phage-ELISA	If-f	dAb	1/11 ^d^	Clone LH7, specific to some breast cancer cell subpopulation
Larsen et al. [70]	1	Cryosections, with shadow stick	PFA, 10 min	During	Rest of the slide	315	On-cell phage-ELISA	IF-f	dAb	1/11 ^d^	Clone LH8, no cross-reaction on healthy breast tissues
Sørensen et al. [71]	1	Cryosections, with shadow stick	Methanol, 5 min	During	Rest of the slide	93	Phage-ELISA on fixed cells	IF-f	dAb then dAb-rFc	1/1	Focus on only one clone
Lykkemark et al. [64]	1	Cryosections, with micropipette dissection	4% PFA, RT, 12 min	During	Rest of the slide	1150 clones; 192 tested	On-cell phage-ELISA	IF-f	dAb	1/1	/

^a^: Perfusion with 0.05% glutaraldehyde, RT, 10 min; ^b^: According to the name of the clones; ^c^: Number of tested clones not provided; ^d^: Eleven clones were tested, one was further studied in each publication; Nb: number; RT: room temperature; mini-Ab: mini-Antibody; dAb: domain Antibody; rFc: rabbit Fc.

**Table 2 antibodies-10-00004-t002:** Antibody target identification.

Authors	Ref.	Target	Techniques
Jakobsen et al., 2007	[28]	GRP78	Yeast two-hybrid screening of a cDNA
Dorfmueller et al., 2016	[32]	ALCAM	Immunoprecipitation + mass spectrometry
Jarutat et al., 2007	[35]	Vimentin	Immunoprecipitation + mass spectrometry
Tanaka et al., 2002	[41]	Actin, Tropomyosin, Actinin, Myosin	Mass spectrometry + cDNA expression library
Ruan et al., 2006	[42]	ALCAM	Sequence similarity with a known anti-ALCAM antibody
Sørensen et al., 2017	[53]	MRPS18A	Protein micro-array

Tanaka et al. developed an “in situ phage screening” laser-assisted selection method [59]. Here, the laser was used to isolate the ROI from the rest of the frozen tissue section before panning that was then carried out on the whole slide. After washes, the previously delimited micro-fragments of interest were transferred into a tube with a micropipette. Phages were then recovered by infection of bacteria or by PCR. PCR recovery gave, as expected, larger numbers of clones than direct rescue. However, no clone was recovered from the smallest 10 × 10 µm^2^ section, and only three from the 33 × 33 µm^2^ section, and they were all non-specific when tested by IHC-f. The authors found that 4800 µm^2^ sections are needed to obtain around 10 clones from direct rescue and more than 70 by PCR. The PCR-based selection protocol was repeated on five samples and a second round performed in two cases, leading to a total of 409 analyzed clones, representing 157 unique sequences. All these clones were tested by IHC-f, and the nine positive clones were further analyzed by Western blotting and target identification: actin (n = 5), myosin heavy chain (n = 2), tropomyosin alpha and actinin-2 (n = 1 clone/each). Only one of the anti-actin clones gave a positive staining by IHC-f on muscle sections. The anti-tropomyosin and anti-actinin-2 clones diffusely stained muscle cells, while the anti-myosin heavy chain clones gave an isoform-dependent staining.

Ruan et al. preselected internalizing scFv with a first selection round on live cells before laser-assisted selection on prostate cancer tissue sections [60,61]. The slides on which the phages were panned were dehydrated before LMD. The authors carried out four antibody selection rounds on FFPE sections and two on cryosections, extracting 20–50 cells/selection. They screened by FACS between 96 and 288 clones per selection and obtained very different positivity rates (from 15% to 88%), even when the Gleason scores of the original tissue sections were similar. The selection rounds on cryosections gave better results (83% of positive clones by FACS vs. 39% with FFPE sections). This could be expected because antigen conformation is less affected in frozen than FFPE tissue sections. A clone from each antibody selection type was tested by IHC. The UA20 clone (selection on FFPE sections) stained specifically the cancer tissue on both FFPE and cryosections, whereas the other clone (selection on frozen sections) stained only cryosections and was less specific.

Ruan et al. observed that phages lose their ability to infect bacteria after LMD. To overcome this major problem, they used PCR amplification to recover the ROI-binding phages [60,61]. Sun et al. [62] and Sørensen et al. [66] attributed this loss of infectivity to the dehydration step, which should not last more than 15 min. Sørensen et al. recommended the use of PBS/glycerol for the last wash to improve phage recovery. However, Sun et al., observed that glycerol preserved infectivity, but prevented the catapulting of the section after LMD. Therefore, they performed LMD before selection, and transferred the catapulted sections in a filter cup in which antibody selection was performed. This strategy presents a major drawback because it eliminates the subtraction step by competition before LMD. The only depletion step performed was in the filter cup, and several types of filter cups were compared to reduce the background noise. After these different adjustments, they performed a unique selection round on approximately 500 cells from 10 catapulted sections from colon cancer cryosections. They obtained 79 clones among which five could stain tumor cells (IF-f screening), but they did not perform any additional characterization. In another study, Sun et al. overcame the dehydration issue by putting a polyethylene naphthalate membrane over the phage-covered tissue section, thus generating a small chamber to keep the tissue moist without impairing the catapulting step [63]. They carried out antibody selection on breast cancer cryosections containing stroma or tumor cells. Approximately 0.5 mm^2^ of panned sections were catapulted, and the bound phages were recovered by bacterial infection. Three panning rounds were performed on the stroma and only one round on tumor cells, and 150 clones/selection were screened by IF-f. Six unique clones identified during the selection on tumor cells gave a specific staining on tumor sections. All positive clones identified during the selection on the tumor stroma (i.e., 20% of the screened phages) contained the same scFv insert (07-2931). This clone strongly stained the tumor stroma by IF-f and by IHC-p. However, when tested on normal tissue samples and on several cancer types, including breast cancer, no particular staining was observed. This suggests that this antibody clone is specific to the individual stroma against which it was selected, limiting its interest for therapeutic and diagnostic applications.

### 3.3. Shadow Stick-Based Antibody Selection

Although LMD allows separating a very small number of cells, this strategy cannot be easily used for single-cell selection. For such applications, Sørensen et al. developed an on-slide selection strategy based on a tool called shadow stick. This method allows selecting antibodies against very rare cells (1 out of several millions), such as circulating tumor cells, for diagnostic and therapeutic purposes [66,67]. Initially proposed for single-cell selection, this method was then transferred to on-tissue selections, using FFPE [65] or frozen sections as targets [69]. The shadow stick is a custom-made tool, composed of a glass rod ending with a minuscule gold disk of 100–120 µm in diameter. Placed above the ROI, it protects the phages bound to this area from UV irradiation after the last panning wash. The ability of phages to replicate in bacteria is impaired by UV-induced DNA damage. Thus, after elution of the whole slide and bacterial infection, only the UV-protected ROI-bound phages can be amplified. For an illustration of the method, the reader can refer to Figure 1 in Larsen et al. [69]. Nevertheless, a compromise is necessary between UV exposure time to remove the background noise, and the risk of losing the infectivity of the relevant phages, because evaporation increases with the exposure time [65]. One of the characteristics of the shadow stick method is to “provide few and highly relevant output clones” [65]. The number of selected clones is low enough to allow their direct screening: eight clones per selection after optimization in 2010 [66]; 12 clones in three selections in 2013 [68]; 40 clones in two selections on FFPE sections in 2015 [65]; 315 clones in 13 selections on frozen sections in 2015 and 2016 [69,70]; and 93 clones in 8 selections on frozen sections in 2017 [71]. This low clone number greatly facilitates the subsequent screening steps.

The precise positioning of the stick over the ROI is ensured by microscopic analysis of the cell morphology or by specific staining. Noteworthy, the method chosen to mark the ROI can significantly affect the target. For instance, PFA fixation before IHC modifies the protein conformation. As discussed in this review, this could be desirable if the aim is to identify antibodies for IHC. Nevertheless, it can be avoided by using two consecutive sections: one for IHC and the other for panning [69,70]. The nature of the targeted protein also can be affected by the marking method. Indeed, pre-treating a cytological preparation for fluorescence in situ hybridization can damage the cells, and thus increase the chances to select antibodies against intracellular proteins [66,68]. This is not desirable when the objective is to identify therapeutic antibodies, but perfectly adequate for diagnostic applications.

In the only published work on shadow stick-based antibody selection on FFPE sections [65], the first selection was carried out on CD31+ blood vessel cells and yielded 27 clones, and the second selection on von Willebrand factor+ cells yielded 13 clones. These 40 clones were first tested by phage-ELISA on PFA-fixed endothelial cells. The four phages with the strongest signal came from the first selection and were further tested by ELISA. Three were produced and purified in the scFv format, and then evaluated as ICC and IHC staining reagent. The patterns observed by IHC-p confirmed the ICC results: clone 2E seemed to bind specifically to endothelial cells, while clone 3B did not provide any staining, and clone 1D was not specific for the cells against which it was selected.

Several shadow stick-based antibody selection experiments were successfully performed on cryosections. For instance, Sørensen et al. targeted CK14^+^/CK19^+^ cells in a breast cancer cryosection [71] and screened the isolated clones first by phage-ELISA on short-term-cultured CK14^+^/CK19^+^ cells, and then by IHC-f as soluble domain antibodies or as dimeric soluble domain antibodies with a rabbit Fc-region. Clone BC5 was also assessed on a frozen tissue micro-array (TMA) composed of 37 breast cancer samples and 3 normal breast samples. Its target was identified as MRPS18A, a mitochondrial protein overexpressed in breast cancer the role of which has not been elucidated yet. In the work by Larsen et al. [69,70], consecutive breast cancer tissue sections were disposed on slides, and CD271^+^ cells were located by IHC in the middle section of the series. The corresponding areas where selected on the consecutive formalin-fixed cryosections for subsequent UV protection with the shadow stick. After overnight incubation of the phages, the last wash before elution was performed with PBS/glycerol to prevent phage dehydration and loss of infectivity. After a first phage-ELISA screening, 35 clones among 315 were further evaluated in ELISA and 11 were tested by IHC-f. Most of them stained the stroma rather than cancer cells. Conversely, clone LH7 stained subsets of breast cancer cells on sections from four patients (basal-like and luminal breast cancer) and did not stain normal breast sections [69]. Clone LH8 stained the tumor cell nests in basal-like and luminal breast cancer samples, with varying intensities [70], but not normal breast sections. Variations in the cancer cell staining intensities of LH7 and LH8 suggest the recognition of antigens that are differentially expressed or modified in some breast cancer subsets.

### 3.4. Micropipette-Assisted Microdissection Strategies

The shadow stick method can be used only when the target cells are very rare and/or clustered on the slide to avoid losing relevant phages by irradiating cells of interest that are not protected by the shadow stick. Larsen et al. suggested that 75–100 cells can be protected by the shadow stick [69,70]. This is similar to the 50-cell sections excised by LMD [60,62]. However, the protection offered by the shadow stick may be less effective at the edges of the shielded area, especially on tissue sections [65] and when the targeted area does not have a circular shape. Irrelevant cells also may be UV-protected, thus introducing a bias and resulting in the selection of non-specific clones. Conversely, if some ROI remain UV-exposed, phages binding to them cannot be retrieved.

To address these limitations, Kristensen’s group proposed to use a micropipette to excise the tissue of interest before phage elution and recover the ROI-binding phages [64]. The first steps are the same as for the shadow stick method: ROI location and phage panning. Then, the ROI is removed with a micropipette and transferred into a tube for the elution step. For an illustration of the method, the reader can refer to Figure 1 in Lykkemark et al. [64]. This method allows the elution of several areas on a given slide, and also prevents the loss of relevant phages fixed on ROI that would not be protected by a single shadow stick. This last point is particularly relevant in the context of on-tissue antibody selection because the cells of interest are not necessarily grouped in a single area, unlike model systems where cells can be manually spiked onto the slide [66]. Micropipette-assisted microdissection is also more versatile than the shadow stick method concerning the ROI dimensions and shape. Thus, this strategy is close to both laser-assisted and shadow stick selections: rare cells of interest (ideally 1/10,000), depletion by competition on the negative tissue, and possibility to discover new targets. Lykkemark et al. demonstrated its efficiency by showing that the elution of two ROI (pericyte-covered capillaries) from a unique human brain tissue cryosection allowed the isolation of 1150 clones [60]. This high output is closer to that of LMD than that of the shadow stick method. After a first screening of 192 phage clones by ELISA on fixed cells, only one (PF9) was retained after additional tests by ELISA and ICC. PF9 is a soluble single-domain antibody that by IHC-f gave a perivascular staining on tissue cryosections from the same samples used for the selection, without recognizing PDGF receptor β and NG2, the two markers generally used for pericyte identification.

## 4. Conclusions and Future Perspectives

Although it is a routine technique, phage display is still rarely used to develop antibodies for tissue section analysis. Currently, most of the available antibodies have been obtained using animal immunization, frequently with synthetic peptides. However, the development of suitable in vitro approaches for antibody discovery might quickly change this, as observed for therapeutic antibodies where display approaches have largely confirmed their value and interest [76].

The results presented in this review indicate that IHC-compatible antibodies can be obtained using phage-displayed naïve antibody libraries. This is true for fresh and frozen tissue samples, and also for FFPE tissue sections. Table 1 summarizes the results presented in this review and shows that IHC-f- and IHC-p-compatible antibodies have been obtained in 16 (73%) and 8 (36%) of the reviewed studies, respectively. However, this high success rate is frequently due to the screening of a large number of clones (Table 1, column “Nb of clones”). In particular for IHC-p, the fraction of positive clones varies widely in the cited publications between 0.13% in Ruan et al. [60] and 31% in Jarutat et al. [53], but with a low median value of 2.4%. The necessity to screen a large number of clones is probably due to the difficulty to set up high-throughput screening procedures by IHC but also to the complexity of tissues. The fraction of positive clones is even lower is we consider as positive only the clones able to stain specifically a collection of positive tissues from different patients and organs, since only few of the antibodies isolated in the reviewed publications were robust enough to set up real IHC applications (Table 1, column “Note”). Of note, the best success rate was obtained by Jarutat et al. [53], who used FFPE sections throughout the procedure, that is for panning, screening, and characterization, illustrating how phage display selections should be conducted to maximize the chance of success.

Thus, three main issues need to be solved before the wide and reliable implementation of phage display for IHC applications. First, IHC-compatible screening methods must be developed to test several hundred of clones (Table 1, column “Nb of clones” for the first screening). Second, the strategies to increase antibody specificity must be improved because many of the isolated clones were not specific enough for diagnostic applications (see text and Table 1, column “Note”). Third, approaches to target specific antigens must be developed because in all studies a cell-targeting approach against unknown antigens was used (Table 2, column “Target”).

Approximately 1/3 of the screening strategies discussed in this review started with a phage ELISA screening performed on cell lysates, fixed cells, or immobilized membranes. However, the reproducibility of on-cell ELISA screening is debatable [71], and some of the positive clones identified by Jensen et al. on a cell monolayer were against secreted proteins rather than cell surface markers [77]. Importantly, many selection methods described here were carried out on tissue samples to ensure clinical relevance, because cultured cells may derive or express some antigens differently due to the lack of microenvironmental stimuli. Therefore, performing the first screening by ELISA on cultured cells is contradictory, as acknowledged by Larsen et al. [65]. ELISA is used for the initial screening only due to the possibility to assess a large number of clones (>3000 by Edwards et al. [47]), but its pertinence to identify antibodies for IHC is limited. On the other hand, the screening performed by ten Haaf et al. [57,58] was meant to identify antibodies against native membrane proteins, but the panning was performed on FFPE sections. This could have led to the selection of antibodies against modified epitopes affected by tissue processing, although this was not tested by the authors. For instance, during antigen retrieval procedures, the high temperatures (90–100 °C) may alter the protein conformation [15], although Shi et al. think that the fixation step protects proteins from denaturation [9].

As summarized by Lipman et al., monoclonal antibodies “should be generated to the state of antigen to which it will eventually need to bind” [78]. Therefore, to identify optimal antibodies for IHC, clones should be selected and also screened on slides. This has been done by few authors who directly assessed phage clones by IHC [45,48,51,52,53,59]. Nevertheless, among them, only Jarutat et al. performed this screening on FFPE tissue samples [53]. Direct on-tissue screening is only possible if the number of output clones is low, because high-throughput screening is not possible by IHC. Indeed, pathological tissues are often limited, and it is not possible, or at least difficult and costly in terms of time and material, to screen hundreds of clones by IHC. Therefore, strategies to reduce the number of slides are needed. Some medium-throughput approaches have been proposed. For instance, Tanaka et al. used LMD to screen up to four clones per slide [59], while Jarutat et al. mounted fast frame grids on TMA slides to form separate chambers and to screen up to six clones on three tissue samples per slide [53].

Tissues constitute a complex source of antigens, the heterogeneity of which can increase the background noise and facilitate the selection of unspecific phages. Therefore, particular care must be taken when choosing the selection strategy, because it may significantly influence the quality and specificity of the isolated antibodies. This is usually accomplished through depletion steps or competition during panning (Table 1, columns “Depletion”) using a comparable tissue. However, a single tissue only presents a subset of the potential targets and background binding is frequently observed when a larger collection of tissues is used during the validation steps. Consequently, subtractive strategies are not always sufficient to prevent the isolation of unspecific phages, as shown by clone 1D isolated by Larsen et al. that recognizes some epithelial cells in addition to endothelial cells, its target [65]. Similarly, the phage-ELISA screening by ten Haaf et al. [58] led to the identification of clones against irrelevant membrane fragments and bovine serum albumin. On the other hand, without subtraction step, Jakobsen et al. successfully selected clone Ab39 which can discriminate various cancer types from normal tissue [46]; this is presumably because pathologies, such as cancer, dramatically change the expression of many proteins.

All the antibody selection experiments described in this review were carried out without any prior knowledge of the targeted antigen. This type of selection can lead to the discovery of new candidate biomarkers or therapeutic targets. As shown in Table 2, after the identification of a specific antibody, the authors frequently used mass spectrometry to identify the targeted antigen. This step can be challenging. Indeed, Sánchez-Martín et al. observed that at least one target was identified in only 55% of the 52 studies they analyzed [41]. In the present review, the target identification rate for tissue antigens was even lower (27%), as the antigens were identified in only 6 studies out of 22. For example, Roovers et al. reported difficulties in performing immunoblot and immunoprecipitation experiments with their antibodies, and hypothesized that they recognized a conformational epitope, sensitive to the modifications induced by fixation [48]. The shadow stick method theoretically allows the discovery of new potential biomarkers; however, the targeted antigen was identified in only one of the six published studies. Finally, an additional drawback of the cell-targeted approach is that most of the identified antigens are highly expressed proteins (Table 2). Therefore, this method is not adequate for identifying antibodies against most proteins in a cell. For instance, the antibodies isolated by Tanaka et al. after a single round of selection without depletion were mostly against the major muscle proteins: myosin, actin, and tropomyosin-α [59]. Myosin and actin account for 65% of the total weight of myofibrillar structural proteins [79]. The rarest target identified, α-actinin 2, represents 2% of all myofibrillar structural proteins, which is still a very high expression level given the fact that the proteome contains more than 1 million proteins (including alternative splicing, post-translational modifications, etc.) [80].

Although IgG is the antibody format used in classical IHC, the IHC-positive phage clones were only rarely tested in this format in the studies discussed in this review (Table 1, column “Antibody format” for immunohistochemical staining), despite the fact that reformatting may affect the clone binding properties [81,82,83]. In the future, screening strategies may include next-generation sequencing (NGS), which has proven successful for identifying rare (0.01%) clones that are difficult to identify using classical screening methods [84]. After NGS, gene synthesis or PCR-based retrieval, the subsequent production of the chosen clones could be performed directly in the IgG format, to rapidly screen them in the final format. Although not fully demonstrated, there is no doubt that coupling new-generation high-quality antibody libraries with NGS-based virtual screening, proper depletion steps, and on-slide selection should allow isolating high-quality antibodies for diagnostic applications and tissue characterization on FFPE sections.

## Figures and Tables

**Figure 1 antibodies-10-00004-f001:**
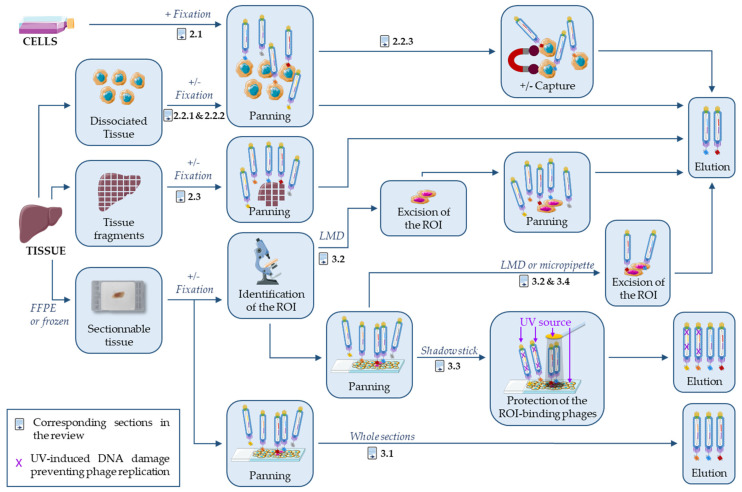
Panning strategies. This figure is a schematic view of the sample preparation and panning steps described in this review. The text sections corresponding to these different strategies are indicated in the figure. Except in one case, the reviewed publications used tissues for panning. Several publications additionally described strategies to select antibodies against particular cells or regions of interest (ROI), using magnetic cell sorting, excision of the ROI with laser-assisted microdissection (LMD) or with a micropipette, and the use of a shadow stick to protect the ROI-binding phages from UV-induced DNA damages. Although the on-slide panning can be carried out before the identification of the ROI, this was not shown here for clarity reasons. Some of the illustrations were obtained from Servier Medical Art library.

## Data Availability

Not applicable.

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
