# Peer review of "Antibody Identification for Antigen Detection in Formalin-Fixed Paraffin-Embedded Tissue Using Phage Display and Naïve Libraries"

_2073-4468, 2021, doi:10.3390/antib10010004_

Round 1

Reviewer 1 Report

This manuscript is a review that focuses on the use of different selection strategies against tissues to identify antibodies using naïve phage libraries. The emphasis is the use of phage display against different formats (e.g. fixed cells, tissues, tissue slides) to identify antibodies that could be suitable for immunohistochemistry, diagnostics, and even therapeutics. The manuscript is well written on a very interesting topic, fits the scope of the journal, and would be of great interest to the readership. There are some comments that should be addressed prior to its acceptance for publication. Questions and comments are listed point-by-point for clarity.

  1. The title should be more reflective that the focus is on formalin/formaldehyde fixed cells and tissues. Initially, from the title and before reading the manuscript, I thought there would have been more review of panning against frozen tissues.

  1. The authors do explicitly state that the emphasis of the review is on identification of antibodies against FFPE-based selections. Can the authors provide additional detail the differences between selection against frozen versus FFPE tissues and highlight the advantages/disadvantages of each? Some information was provided in the introduction, but additional detail would make clearer and more evident of the differences (in the reader’s mind).

  1. Lines 73-74 “Transfected cell lines…”. While I understand what the authors are trying to state, it may not be clear to reader who is less experienced in the area. Please add additional language to make clearer that is referring to a single cell type that is +/- transfected with the target that allows for positive/negative selection, and therefore the only difference is the target.

  1. In general, can the authors comment on why with selections, often only a fraction of the selected clones actually work? It would provide good perspective to the reader.

  1. Throughout the manuscript, the authors often only refer to one or a few references (for example, in section 2.1, only Gur et al is described). Please go back and provide more references wherever possible to give as much literature review as possible.

  1. There is some confusion--if the focus is on fixed cells/tissues, why discuss freshly dissociated tissues in section 2.2.1?

Reviewer 2 Report

In this review, the authors summarized the kinds of examples of antibody selection on fixed or unfixed cells, tissues, and sections using phage-displayed libraries. It will be very helpful to the people who want to pan against some unknown antigens on cells or tissues. Only one suggestion: it would be perfect if the authors could add some schematics to show the processes of shadow stick-based antibody selection or other selections.

Reviewer 3 Report

I rarely review papers as complete and flawless as this one. The review process was thorough; references not only cited, but also criticized; information beautifully summarized on the table. Text very easy to read/follow, with incredible style. I couldn't find a single thing to comment/correct. Bravo.

Reviewer 4 Report

The authors provided a well-documented overview of the impact of phage display technology in the identification of tissue-specific antibodies. In addition to this, with this article, they provide an interesting starting point for a discussion on the next challenges for the phage--passionated researchers. The article is well written but It could be improved by:
1) (lines 27-29): explain in more detail the antigen masking phenomenon.
2) (lines 45-46): phage display is a dynamic field of research and an increased number of contributions have been published after the 2018 Nobel award (PMID: 30606501 is just an example).
3) (lines 46-48): Phage display technology is not limited to antibodies discovery providing also an impressive impact in the field of peptides and peptidomimetics. Please include a little section focused on this aspect (PMID: 22754323; PMID: 31718942; PMID: 22754323; are just an example of references that can help you)
2) (line 69): provide a little introduction to the section and tab.

I hope to soon receive the revisioned version of the paper,

good luck!

Round 2

Reviewer 4 Report

Thanks to the author's efforts